# Production of Drinking Water with Membranes with Simultaneous Utilization of Concentrate and Reject Effluent after Sludge Dewatering

**DOI:** 10.3390/membranes13020133

**Published:** 2023-01-19

**Authors:** Alexei Pervov, Dmitry Spitsov

**Affiliations:** Department of Water Supply, Moscow State University of Civil Engineering, 26, Yaroslaskoye Highway, 129337 Moscow, Russia

**Keywords:** reverse osmosis, nanofiltration, sludge dewatering, sludge moisture, pretreatment, membrane fouling, concentrate utilization

## Abstract

A new technology is described that enables us to completely exclude liquid discharges during production of drinking water from surface sources. The proposed described technological scheme separates the natural water into a stream of purified drinking water and dewatered sludge. The sludge moisture has a value of 80 percent. The experimental program is described to treat the natural water with nanofiltration membranes and to produce a drinking-quality water with recovery value of 0.99 and higher. Concentrate of membrane plant is mixed with the wet sludge and the reject effluent after sludge dewatering is again treated by reverse osmosis membranes and returned back to the sludge thickening tank. Results of experiments to treat reject water after sludge dewatering are presented. The use of nanofiltration membranes provides reduction in the Total Dissolved Solids content (TDS), aluminum, color and oxidation to meet drinking water standards. Experimental plots are presented that can be used to select membrane characteristics and to predict product water chemical composition at each stage of the membrane treatment scheme. Concentrate of membrane treatment plant is mixed with the wet sludge in the thickening tank. The sludge, after the thickening tank, is dewatered using either filter-press or centrifugal equipment. The reject (or fugate), after sludge dewatering, is treated by membrane facility to separate it into deionized water stream and concentrate stream. The deionized water can be mixed with the feed water or drinking water and the concentrate stream is returned back to the thickening tank. Thus, the salt balance is maintained in the thickening tank, whereby all dissolved salts and impurities that are rejected by membranes are collected in the thickening tank, and then are withdrawn together with the dewatered sludge. Based on the results of experimental data processing, balance diagrams of the sludge dehydration process with waste water purification at the membrane plant and with the addition of the membrane plant concentrate to the sludge thickener are presented, according to which all contaminants removed by the membranes are removed together with the sludge.

## 1. Introduction

The use of reverse osmosis membrane technology for the purification of natural and waste water has been carried out over the past 40 years [1,2,3,4,5,6]. In recent years, interest has increased in nanofiltration as a method for preparing drinking water from surface water sources [1,2], due to the high efficiency of retention by these membranes of organic substances that form water color [7,8,9,10]. Due to the “universality” of membranes in retaining organic substances of various natures and molecular weights, reverse osmosis and nanofiltration membranes are already widely used in the processes of post-treatment of domestic waste water for the purpose of their reuse [11,12,13,14]. In addition, due to the different selectivity of membranes (retention efficiency) for monovalent and multivalent ions, membranes are used in the processes of industrial waste water treatment [7,8,9,10,11,12,13,14,15]. Finally, thanks to the development of nanofiltration membranes, it became possible to deepen multiple concentrations of waste brines, effluents and concentrates for the purpose of their further utilization. In particular, there is a lot of information on the use of nanofiltration for purification of centrifuges after dehydration of mineralized sediments of natural and waste waters [16,17].

To treat water with high color, additionally, ozonation is used to oxidize and destruct organic substances with subsequent sorption. Moreover, foulants of anthropogenic origin, such as detergent, pesticides and halogenocarbons, are also reduced using ozonation and sorption techniques [6]. The real breakthrough in the quality drinking water production was the combination of ozone, sorption and membrane techniques. In these technological schemes, ultrafiltration membranes are used, but membranes in these schemes are assigned only a role of a fine filter to remove fragments of coal particles and pathogenic bacteria living in filtration media [6]. Moreover, the high color and high water oxidation that is attributed to halogenocarbons and other volatile organics can be successfully solved using nanofiltration membranes [1,2,3,4]. Application of nanofiltration membranes does not require the use of sorption materials that require high operational costs [5]. This is confirmed by long successful experience of a number of large-scale nanofiltration facilities [1,2]. The main disadvantage of the application of nanofiltration and reverse osmosis membranes at large drinking water supply installations is the concentrate disposal problem [2,3,17]. The problem of concentrate reduction and utilization at water supply plants is still not completely solved [17], mainly due to the presence of low soluble salts in concentrate and high cost of their removal [17,18].

There are a number of reverse osmosis concentrate reduction techniques that are used for “zero discharge” in technical water production schemes. However, these tools require high reagents and energy consumption to remove hardness, further increase recovery and evaporate concentrate [17]. In this article, authors investigated a new approach to the use of nanofiltration and elimination of concentrate discharges. The idea to “hide” concentrate in the dewatered sludge occurred after authors conducted research to treat reject effluents after sludge dewatering. When surface water is treated by nanofiltration, a pretreatment is required to remove suspended and colloidal matter, as well as color, from the feed water (Figure 1). The conventional pretreatment scheme includes usually clarifiers and rapid filters (Figure 1a). Moreover, ultrafiltration is successfully used for color and turbidity removal (Figure 1b). In both cases, sedimentation sludge after filter backwashes and clarification is collected and dewatered. However, the dewatered sludge moisture usually has a value of 80 percent. Moreover, when the sludge is dewatered using centrifuges of filter-press equipment, a fugate or reject effluent is formed with a volume of 2 percent of the total volume of purified water. In our previous research, we have attempted to treat and utilize the reject water discharges using reverse osmosis with disposal of rejected impurities together with dewatered sludge [19].

Figure 2 describes principles of reject effluent treatment with reverse osmosis membranes. The reject water after sludge dewatering is treated by reverse osmosis membrane facility and separated into product water and concentrate. Product water quality enables us to add it to the drinking water flow, as membranes efficiently remove organics and aluminum from the reject stream. Concentrate is returned back to the sludge thickening tank and mixed with the fresh wet sludge. The required TDS value is maintained in the sludge thickener to satisfy equilibrium conditions: the amount of salts entering the sludge thickening tank should be equal to the amount of salts withdrawn from the tank, together with dewatered sludge. Assuming the feed water flow rate of 1000 cubic meters per hour, the reject water flow rate can be 20 cubic meter per hour and the flow withdrawn from the sludge as a sludge moisture can be 4 cubic meters per hour. These data are taken according to the average values of parameters, calculated for Moskva River water treatment. To maintain salt balance during sludge dewatering, the reject water should be treated and concentrated by 5–10 times (Figure 2a).

The purpose of this work was to study the possibility of using membrane technologies (nanofiltration and reverse osmosis), both for natural water purification (to remove excessive color, to reduce oxidation and to remove anthropogenic contaminants, such as halogenocarbons and pesticides) as well as to demonstrate the possibility to utilize concentrate together with dewatered sludge.

This article presents the results of the research conducted at the department of Water supply and sewage of Moscow State University of Civil Engineering to develop a new technique to treat surface water and to utilize concentrate of a membrane plant [19,20,21,22]. This approach involves reduction in concentrate flow and mixing it with the wet sludge forwarded to the sludge thickening tank prior to its dewatering (Figure 2b). Thus, all rejected salts and impurities are discharged together with dewatered sludge as a sludge moisture [19]. The salt and flow balance diagram of the proposed process is presented on Figure 2b.

Principles of balance calculation of flows and concentrations shown in the scheme of Figure 3 are based on the observance of the material balance. When dehydrating sludge from waterworks, centrifuge consumption is about 2% of the source water consumption, and the consumption of water removed along with the sludge (which is the moisture content of the sludge) is 0.4% of the source water consumption. In the example under consideration, the consumption of initial water is 1000 cubic meters per hour, respectively, the consumption of waste water is 20 cubic meters per hour, and the consumption of water removed with sediment is 4 cubic meters. In order for all contaminants removed during the cleaning process to be removed with sediment, it is necessary to maintain a material balance: the amount of salt sent to the dehydration unit along with the concentrate must be equal to the amount of salt removed along with the sediment. Thus, at a concentrate flow rate of 4 cubic meters and with its total salt content of 25 g per liter, the total amount of salts removed from the reverse osmosis unit per hour will be 100 kg. The same amount of salts should be removed with water in the sediment. At a flow rate of 4 cubic meters per hour of water withdrawn together with the sludge, its salinity should also have a value of 25 g per liter. In addition, the value of the total salt content of water coming from the thickener to the centrifuge should also be 50 g per liter. The thickener receives two streams: 4 cubic meters per hour of concentrate and 24 cubic meters per hour of sludge with a total salt content of 500 mg/L. Therefore, the reverse osmosis unit used in the circulation system must separate the mixture of these streams (4 plus 24) into a concentrate with a salinity of 50 g per liter (at a flow rate of 4 cubic meters per hour) and permeate (24 cubic meters per hour) with a salinity of 500 mg/L (Figure 2b).

To provide a reduction in feed water flow rate by 100–200 times and to reach TDS value of 35,000–50,000 ppm in concentrate stream, a “cascade” of nanofiltration membranes is used (Figure 3). Principles of concentration increase using low rejection membranes are shown in Figure 3. Concentrate from the first-stage membrane element enters the second-stage membrane element, and then concentrate of the second-stage membrane element enters the third-stage membrane element. Product water of the first-stage membranes is a drinking-quality water that the membrane plant produces. Usually, the recovery value of the first stage can be 0.75–0.85. Concentrate after the first-stage membrane enters the second-stage membrane. The TDS value of the second-stage product water is similar to the feed water TDS; therefore, the product flow of the second stage is forwarded to the feed water inlet. The recovery value of the second membrane stage can also lie between 0.8 and 0.85. Thus, second-stage concentrate flow rate can equal 1/30–1/40 of the feed water flow rate. Similarly, as the third-stage permeate TDS value is similar to the first-stage concentrate TDS, the third-stage product water is forwarded to the second-stage inlet. The third-stage recovery value is also 0.8; thus, the feed water flow is reduced by 180–200 times and concentrate TDS can reach 50,000 ppm value. This principle is used in brine concentration systems [21,22] and provides high concentration values at comparatively low power consumption. In our experiments, we reached 50,000 ppm TDS values applying pressure of 16 Bars.

To treat reject water after dewatering (Figure 2b), a double-stage membrane facility should be used [22]. The reject water in our case can have a high TDS value of 50,000 ppm. Moreover, reject effluents contain aluminum and COD. The double-stage flow diagram is presented on Figure 4. The first stage uses low rejection nanofiltration membranes to reach the required TDS value of 50,000 ppm at a recovery value of 0.5. The product water of nanofiltration membrane (of the first stage) is further treated by the low-pressure reverse osmosis membrane used on the second stage (Figure 4) to reach a low TDS value and remove main impurities (such as aluminum and organics). The second-stage concentrate is forwarded to the inlet of the first stage (Figure 4).

The use of reverse osmosis membrane devices at the second stage of purification makes it possible to reduce the concentration of organic substances in the water treated at the second stage and increase the productivity of purified water [22]. This article aims at investigating the process of concentrate flow reduction to reach a concentrate value 100–200 times less than the feed water flow and evaluating the main technical and operation parameters of the process. These parameters include concentrate and product water composition prognosis, product flow and rejection characteristics of membranes on each stage of the scheme, scaling and fouling membrane characteristics.

## 2. Experimental Part: Materials and Equipment

The experimental test program included four steps:Treatment of natural water from the surface intake with nanofiltration membranes to evaluate the maximum recovery that provides the required product water quality.Reduction in concentrate volume by 100–200 times as compared to the feed water volume and evaluation of the influence of the water-concentrating process on membrane flux decrease and product quality deterioration.Evaluation of scaling rates in membrane modules throughout the concentrating cycles.Treatment of reject effluent after sludge dewatering to produce quality drinking water and evaluation of membrane efficiencies on each membrane stage.

In experiments on the first stage, rejection of main water components was investigated as well as the influence of the feed water-concentrating process on the membrane operational characteristics. To concentrate feed water by 200 times, two steps of testing were used. As the first step, an initial volume of 200 L was concentrated by 10 times and feed water volume was reduced to a value of 20 L. As the second step, the concentrate volume of 20 L was reduced by 10 times to reach a concentrate volume value of 1 L. In these experiments, the Moscow tap water was used to investigate the increase in TDS and concentrations of different ions: calcium, chlorides, bicarbonates and sulphates. Moreover, as a third step, concentrating of the reject after sludge dewatering by 10 times using a nanofiltration membrane was arranged. The reject water volume was 20 L and taken from Moscow water treatment plant. Concentration values of calcium, chlorides, bicarbonates and sulphates had the same value as in the tap water. However, the reject effluent is distinguished by high concentrations of organics and aluminum. The color values in this effluent can reach 70–80 degrees NTU, aluminum concentration was 30 ppm and oxidation was 35 ppm. The chemical oxygen demand value (TOC) was 50 ppm. As well as the above-mentioned impurities, reject contains manganese, nickel, mercury, lead and arsenic in quantities tens of times exceeding the permissible regulation limits. Therefore, the sludge after dewatering is forwarded to landfill and the reject stream is discharged into the sanitation sewer to join municipal waste water.

The experimental procedure consisted of the circulation of the feed water in the system with constant product water withdrawal to simulate the process of water separation in membrane modules. The feed water was added to feed water tank 1 (Figure 5) from which water is pumped by device 2 to the membrane module 3. 

In the membrane module, the test solution was separated into permeate (purified water) and concentrate. The concentrate was returned to tank 1 and permeate was collected in tank 4. The Procon rotary pump was used, producing the flow value of 180–200 L per hour at a pressure of 16 bar. The experiments were carried out using commercial membrane elements of the 1812 standard model produced by “Toray advanced Materials Korea Inc.” (the manufacturer of CSM membrane technologies). Membrane elements tailored with low-pressure reverse osmosis BLN membranes (salt rejection value of 95–96%), and also with nanofiltration 70 NE membranes (salt rejection value of 70%). The membrane surface area in membrane elements of 182 standard was 0.5 square meters. Moreover, values of membrane rejection of different impurities and membrane flux values as dependencies on the initial volume reduction coefficient value (K) were determined on each stage of membrane treatment. The coefficient K is defined as the ratio of initial feed water volume Vf in tank 1 to the volume of concentrate at the different moment of time during experiment conductance Vt (Vf/Vt). This initial flow reduction coefficient K is an important indicator of the membrane plant performance and, in industrial conditions, is also defined as the ratio of the feed water flow rate to the concentrate flow rate (Qf/Qc) and, by its purpose, it is close to such an indicator as recovery, which is the ratio of the product flow rate to the feed water flow rate (Qp/Qf) and is described by the equation: Qp/Qf = 11/K. On the first stage of our test program, an element with nanofiltration membranes was used. Results of experiments devoted to evaluation of membrane rejection characteristics are presented in Figure 6, Figure 7 and Figure 8.

Concentrate after the first stage was forwarded to a further volume reduction stage. Concentrate composition (concentration values of calcium, chloride and sulphate) as well as COD values in the end of the first series are shown in Figure 6a and correspond to K value that equals 6. This concentrate was further treated in the second series and was used as feed water. The values of concentrations of all ingredients are shown in Figure 6b as functions of K.

The second stage of our test program was devoted to evaluation of scaling rates that can sufficiently reduce the efficiency of the technology proposed. Determination of the scaling rates was carried out in accordance to the test procedure developed previously by the authors and described in a number of publications [17]. During experimental test runs (steps 1 and 2), calcium concentration values were evaluated at different feed-water-concentrating steps as dependencies on K (Figure 9). 

During each test run, we simultaneously evaluated concentrations of calcium and other components, both in tank 1 and tank 4 (Figure 5). In case scaling and calcium carbonate accumulation takes place in the membrane module, for different K values, we can evaluate the amount of accumulated calcium carbonate coming down for reasons of material balance according to Equation (1):M = V_c_ × C_c_ − V_p_ × C_p_(1)
where M is amount of accumulated calcium carbonate, milliequivalents; V_c_ and V_p_, respectively, are volumes in tank 1 and tank 4; C_c_ and C_p_ are, respectively, calcium concentration values in concentrate tank 1 and permeate tank 2, milliequivalents per liter. The obtained amounts of deposited calcium can be presented as dependencies on time to draw plots of M values versus time T. Thus, scaling rates can be determined as tangent values of these dependencies. These obtained dependencies of scaling rate values on K values are presented in Figure 9.

Evaluation of calcium carbonate growth rates was implemented during concentration experiments at both steps 1 and 2 on stage 1. It is well known that scaling always takes place in membrane modules during natural water treatment and the main measure to control scaling is antiscalant addition [17]. Therefore, we added antiscalant “Aminat-K” [17] to the feed water with a dose of 3 ppm.

On the third stage of the experimental program, the process of the reject effluent treatment with nanofiltration and reverse osmosis membranes was investigated. According to the flow diagram presented in Figure 2b, the mixture of the water after sludge thickening and membrane concentrate from the thickening tank is collected and treated by membranes. This water flow should be reduced by 2–2.5 times and separated to produce a quality water and concentrate to be returned back to the sludge thickener. The concentrate TDS should reach the value of 25,000 ppm, equal to the TDS value of concentrate produced by nanofiltration plant (Figure 2b). During water-concentrating experiments, we have collected the data of concentrate and product water on all stages of concentrating. The corresponding product water volume of two liters was collected during the second step conductance after the volume in tank 4 reached 17 L. Then, these two liters were treated using the same procedure (Figure 5) using a membrane element with reverse osmosis membranes. A small RO-900 pump (80 L per hour/8 Bars) pump was used in this series of experiments. The chemical composition of the concentrate and permeate of nanofiltration membranes was already investigated and presented in Figure 8.

## 3. Discussion of Experimental Results

The results of the experiments presented in Figure 6 and Figure 7 show the concentrations of contaminants in product water and concentrate during the natural water concentrating throughout the first series of experiments.

Results of the reject effluent treatment using nanofiltration membranes are also presented as dependencies of concentration values of main impurities in product water and concentrate on the K values reached during the test solution concentrating (Figure 8).

The processing of the obtained results enabled us to develop the dependencies of membrane rejection characteristics for different impurities in the form of logarithmic functions, as shown in Figure 9.

Figure 10 demonstrates results of calcium carbonate growth rate evaluation for different conditions. Comparison is provided of scaling in nanofiltration and reverse osmosis membrane modules using antiscalant dosing and without antiscalant addition. The comparative study was conducted on the first step of the first experimental series. As can be seen in Figure 10, scaling in nanofiltration membrane modules is substantially lower than in reverse osmosis modules. In the second step of feed water concentrating, calcium concentrations in tank 1 (Figure 5) were determined (Figure 1a) and calcium carbonate growth rates were evaluated by the end of this test run. As can be seen in Figure 10b, scaling rate values obtained throughout the whole concentrating cycle do not exceed the scaling rates in nanofiltration modules that occur during the first step of the experimental program when the feed water was concentrated only by 10 times. This can be explained by high salt concentration increase during water concentrating and reduction in super saturation rate of calcium carbonate.

To develop the membrane facility to treat the reject stream after sludge dewatering (Figure 4), a new test cycle was organized. Concentrate after the first series of experiments (1 L) was mixed with 1 L of the fugate (reject water) collected at the water purification plant in Moscow. In conformity with the mass balance shown in Figure 2b, this mixture was concentrated by two times to reach concentrate TDS of 25,000 ppm. The purpose of this experiment was evaluation of membrane flux on both stages of the double-stage membrane facility as well as evaluation of membrane rejections of aluminum and COD. Figure 7, Figure 8, Figure 9 and Figure 10 demonstrate that it can be reached, as we collect all impurities and put them into the water that is withdrawn with the dewatered sludge. Figure 11 shows dependencies of the specific product flow of nanofiltration membrane on the first stage and of reverse osmosis membrane on the second stage on K. Dependencies of all ionic species contained in water and COD in concentrate (a) and product water (b) on K values for nanofiltration and reverse osmosis membranes, respectively, are presented in Figure 12 and Figure 13.

The described approach to utilize concentrate provides a substantial economic effect. It is well known that, to treat water from surface sources with high permanganate oxidation values (over 10–12 mg/L), ozone and sorption techniques are applied. Addition of powder activated carbon of filtration through granular activated carbon bed results in high operational costs, as activated carbon has a limited adsorption ability and high cost that varies from USD 5000 to 10,000 per one ton of the product. Nanofiltration membranes reject volatile organics, halogenocarbons and pesticides (with equivalent weight that varies from 100 to 300 g per equivalent) without the use of consumables. For comparison of cost of materials purchased during one year of operation of a water treatment station with 1000 cubic meter per hour capacity, we evaluate the annual cost of activated carbon dose of 10 mg per liter ranging from USD 350,000 to 700,000 and annual costs for membrane replacement as USD 100,000–120,000. One of the main economical reasons to refrain from using membranes in water treatment practice is the existence of concentrate discharges that can equal 15–30 percent of the total feed water flow. Usually, as it is presented in Figure 2b, conventional water treatment plants discharge into the sanitary sewer about 2 percent of the total feed water flow as a reject (or fugate) after sludge dewatering. For the example discussed in the present article and shown in Figure 2b, for 1000 cubic meters per hour capacity, assuming the discharge payment of USD 0.5 per one cubic meter, the annual payment for discharge of fugate can be about USD 70,000. This amount is much lower than the annual payment for membrane replacement of the membrane facility used to treat and concentrate fugate and to provide concentrate flow reduction from 100 to 4 cubic meters per hour, which can be estimated as USD 12,000 per year.

## 4. Conclusions

Application of nanofiltration method to produce drinking water for surface intakes requires the solution for the concentrate disposal problem. A new approach is proposed to “hide” concentrate in the sludge during the sludge dewatering process and withdraw it together with dewatered sludge.Based on the results of experimental data processing, balance flow diagrams of the sludge dewatering process are presented.The use of nanofiltration and reverse osmosis membranes enables us to produce a quality drinking water, both from the natural water from the surface intakes and from the sludge dewatering reject stream. Thus, a “zero discharge” effect can be reached during drinking water production.

## Figures and Tables

**Figure 1 membranes-13-00133-f001:**
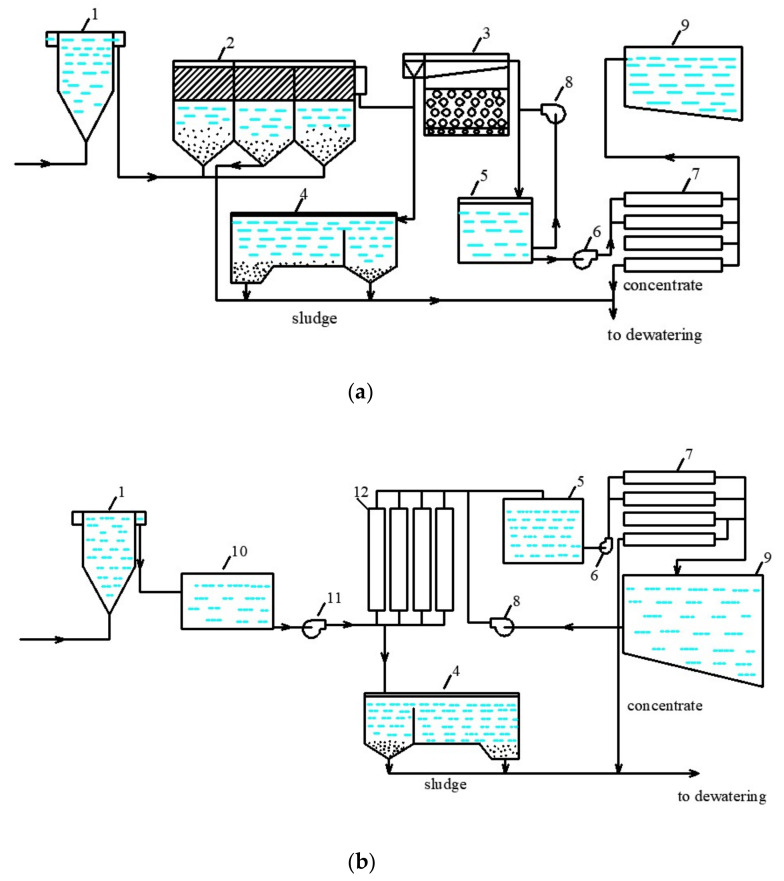
Flow diagrams of drinking-quality water production using nanofiltration membrane filter backwash water reuse and sludge disposal: (**a**)—water treatment plants that use conventional clarification process for pretreatment; (**b**)—water treatment plants that use ultrafiltration for pretreatment. 1—mixer; 2—clarifier with a sludge blanket and lamels; 3—rapid filter; 4—sludge thickener; 5—intermediate reservoir; 6—working pump of the reverse osmosis unit; 7—reverse osmosis membrane modules; 8—back washing pumps; 9—clarified water tank; 10—flocculation chamber; 11—working pump of the ultrafiltration membrane modules; 12—ultrafiltration membrane modules.

**Figure 2 membranes-13-00133-f002:**
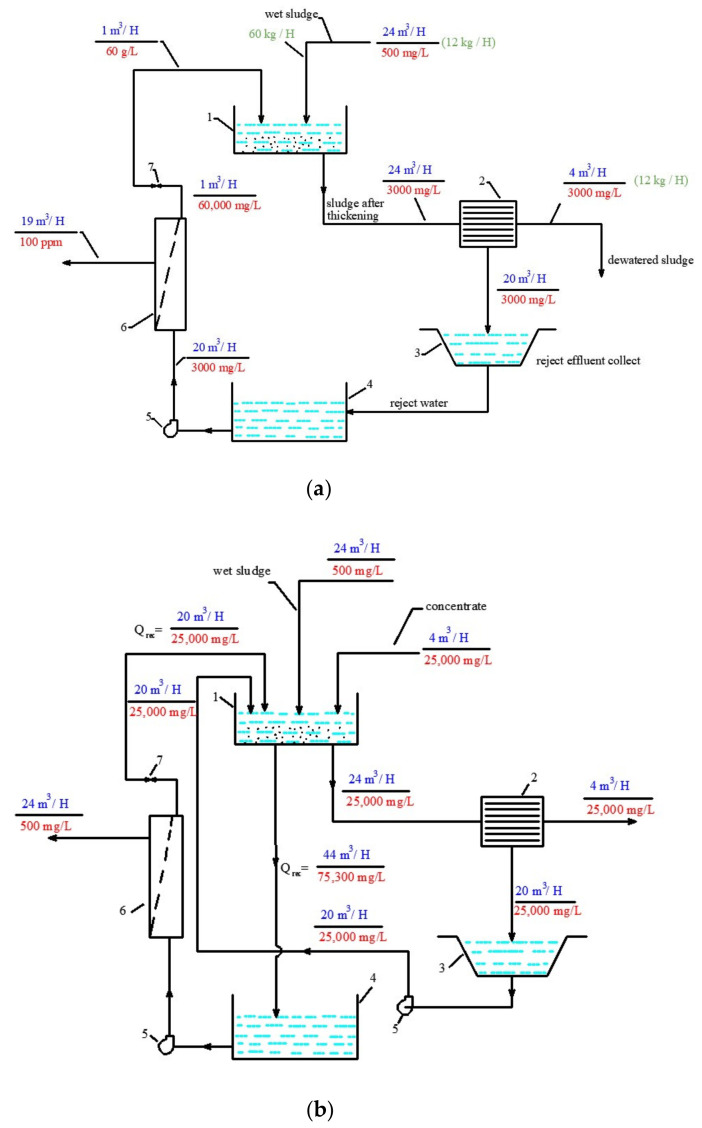
Water and dissolved salts balance flow diagrams of sludge dewatering, reject (fugate) and membrane concentrate disposal and utilization: (**a**) treatment of fugate after sludge dewatering using reverse osmosis and concentrate withdrawal together with dewatered sludge; (**b**) utilization of nanofiltration membrane plant concentrate together with dewatered sludge. 1—sludge thickener; 2—centrifuge; reject effluent collection tank; 3—intermediate tank; 5—working pump of reverse osmosis facility; 6—reverse osmosis membrane modules; 7—pressure regulation valve.

**Figure 3 membranes-13-00133-f003:**
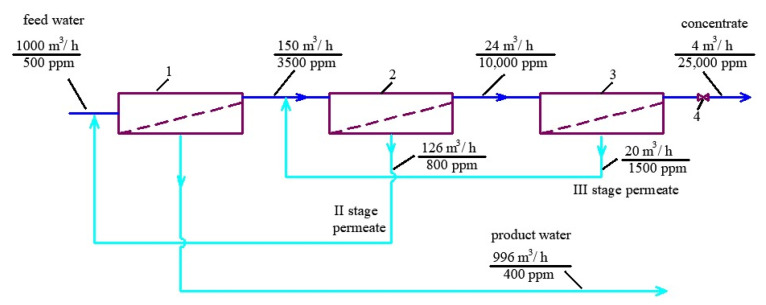
Principles of reducing the volume of concentrate by 100–200 times during natural water treatment using low rejection nanofiltration membranes: 1—the first-stage membrane module; 2—the second-stage membrane module; 3—the third stage membrane module.

**Figure 4 membranes-13-00133-f004:**
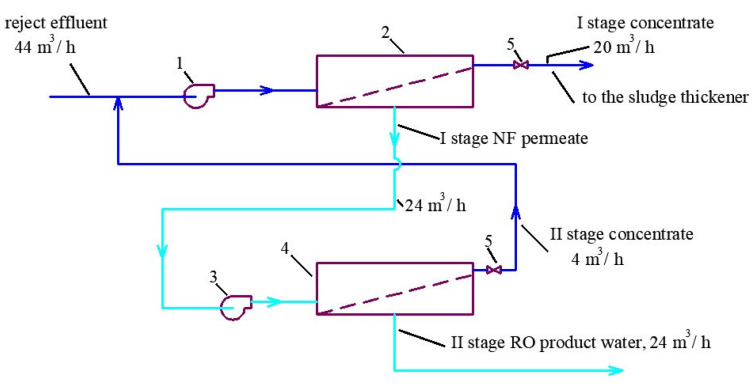
A flow diagram of the double-stage membrane facility to treat mineralized effluent after sludge dewatering: 1—working pump for the first-stage membranes (nanofiltration); 2—nanofiltration membrane module; 3—working pump for the second-stage reverse osmosis membranes; 4—reverse osmosis membrane modules; 5—pressure regulation valve.

**Figure 5 membranes-13-00133-f005:**
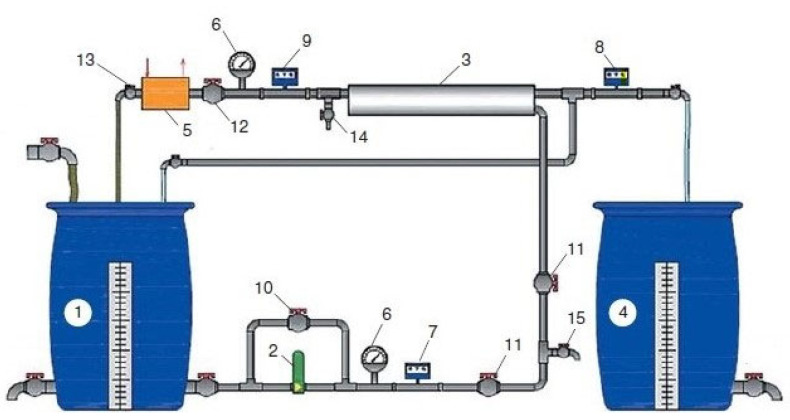
Experimental test membrane unit flow diagram: 1—tank for the feed water; 2—working pump; 3—membrane element in the pressure vessel; 4—permeate collection; 5—heat exchanger; 6—manometer; 7–9—flow meters; 10—bypass valve; 11—flow and pressure regulation valve; 12—pressure regulation valve; 13—valve for adjusting the flow of cooling water; 14, 15—samplers.

**Figure 6 membranes-13-00133-f006:**
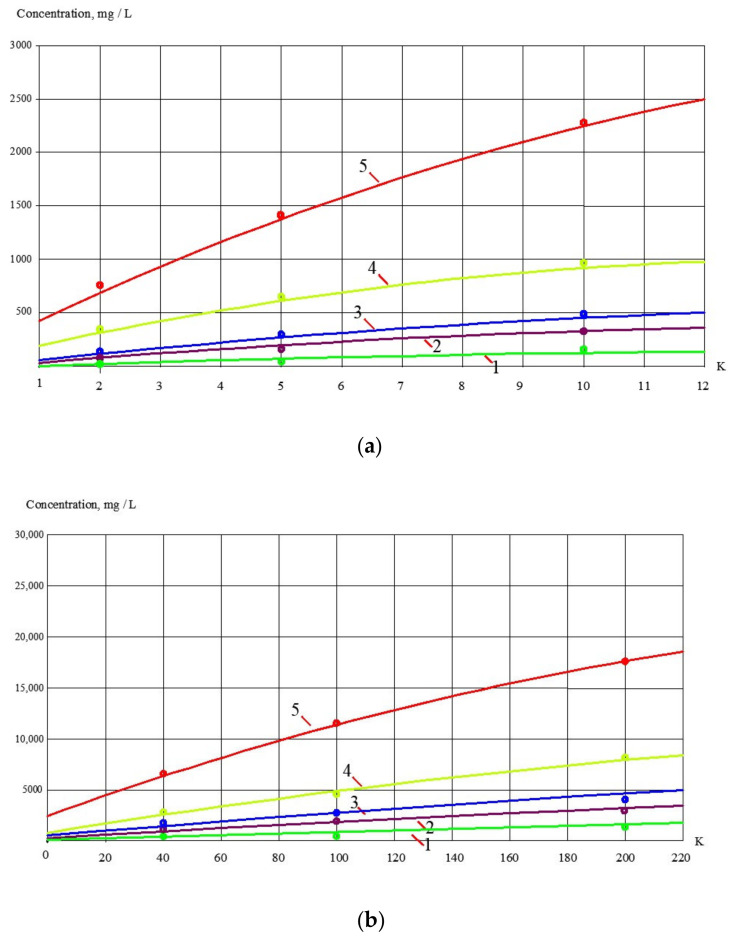
Dependences of calcium, chloride, sulphate and bicarbonate ion concentrations, as well as TDS values on K in the nanofiltration membrane concentrate at the first step (**a**) and on the second step (**b**) of the first series of natural feed water concentrating: 1—chlorides; 2—sulphates; 3—calcium; 4—bicarbonates; 5—TDS.

**Figure 7 membranes-13-00133-f007:**
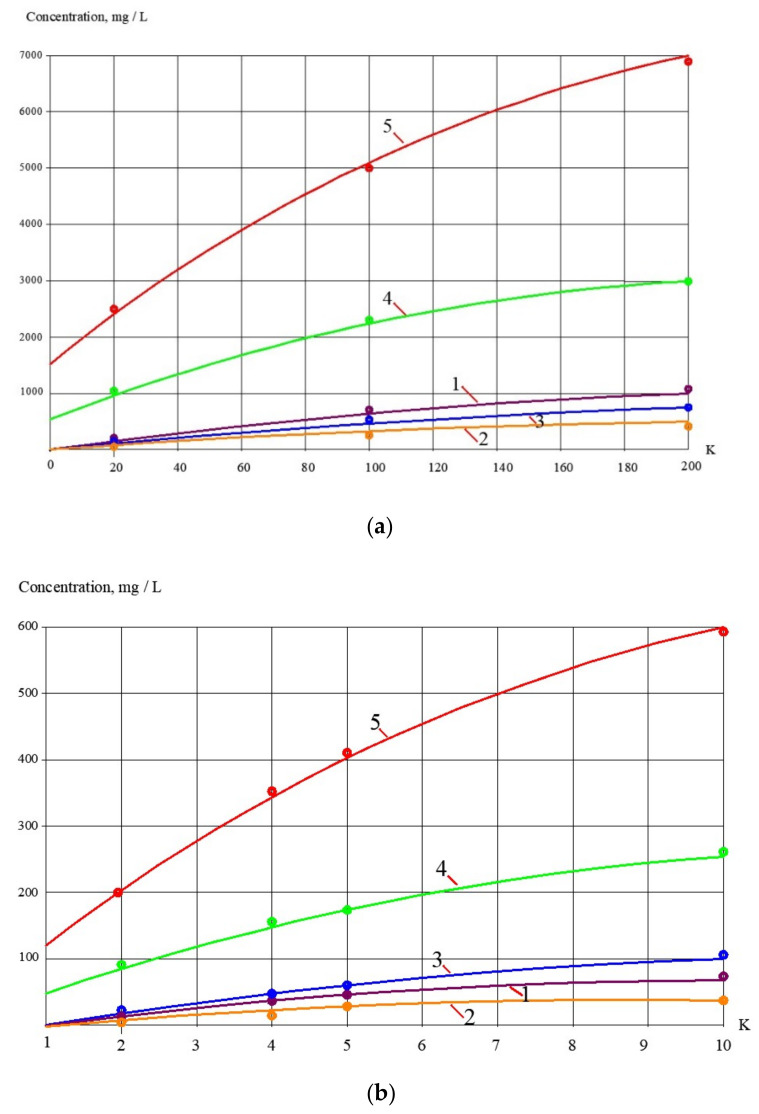
Dependences of calcium, chloride, sulphate and bicarbonate ions, as well as TDS values on K in the nanofiltration membrane permeate at the first step (**a**) and on the second step (**b**) of the first series of natural feed water concentrating: 1—chloride; 2—sulphates; 3—calcium; 4—bicarbonates; 5—TDS.

**Figure 8 membranes-13-00133-f008:**
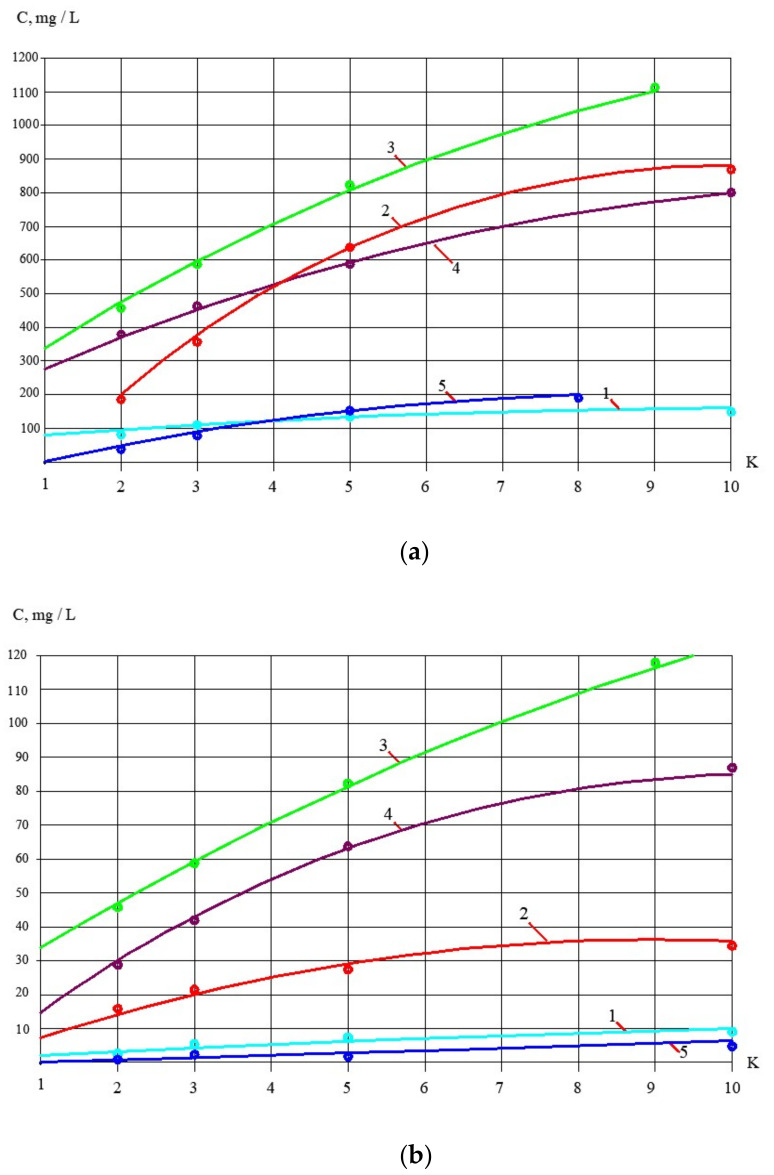
Results of treatment of the reject effluent after sludge dewatering with nanofiltration membrane (step 3, the first series of experiments). Dependencies of different impurity concentrations on K values: **a **– in product water; **b** – in concentrate. 1—chlorides; 2—sulphates; 3—calcium; 4—COD; 5—aluminum.

**Figure 9 membranes-13-00133-f009:**
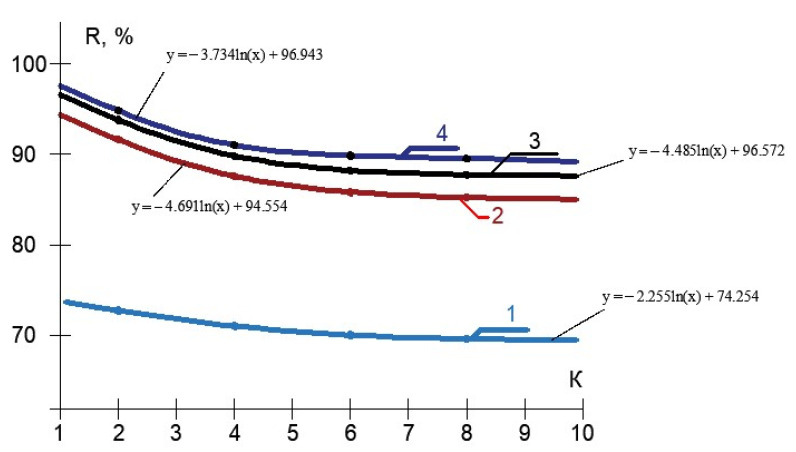
Results of experimental data processing: dependencies of nanofiltration membrane rejection values of different ions and impurities on K in the form of a natural logarithmic functions: 1—chlorides; 2—COD; 3—aluminum; 4—sulphates.

**Figure 10 membranes-13-00133-f010:**
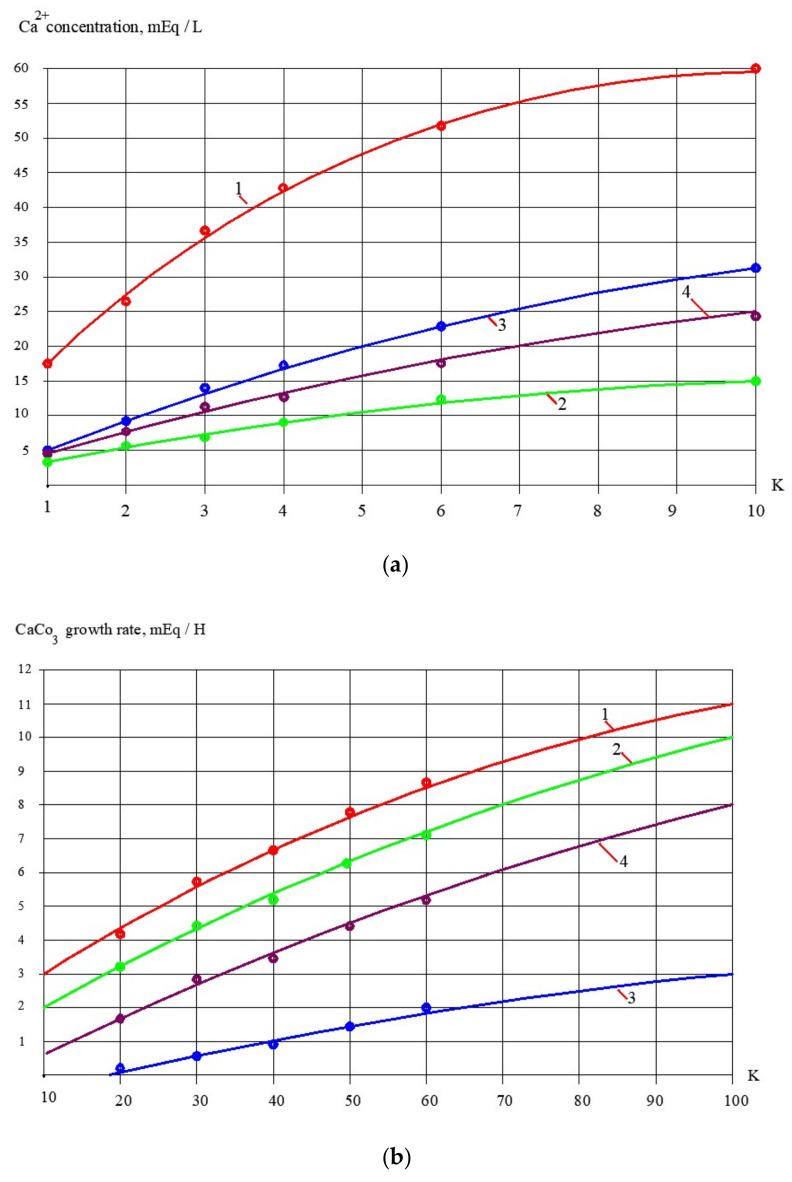
Results of calcium carbonate growth rate evaluations in the first series when natural water volume was reduced by 200 times: (**a**) dependencies of calcium concentration values on T during natural water concentration experiments; (**b**) dependencies of calcium carbonate growth rate values on K determined under different conditions: 1—concentrating of the feed water on the second step of the first experimental series (concentrating of the water concentrate previously concentrated by 10 times at the first step, using “Aminat-K”); 2—concentrating of the natural water by 10 times at the second step of the first experimental series using nanofiltration membrane, without antiscalant addition; 3—concentrating of the natural feed water at the first step of the first series using nanofiltration membrane, “Aminat-K” dosing, 3 ppm; 4—concentrating of feed water by 10 times at the second stage using reverse osmosis membrane, “Aminat-K” dosing, 3 ppm.

**Figure 11 membranes-13-00133-f011:**
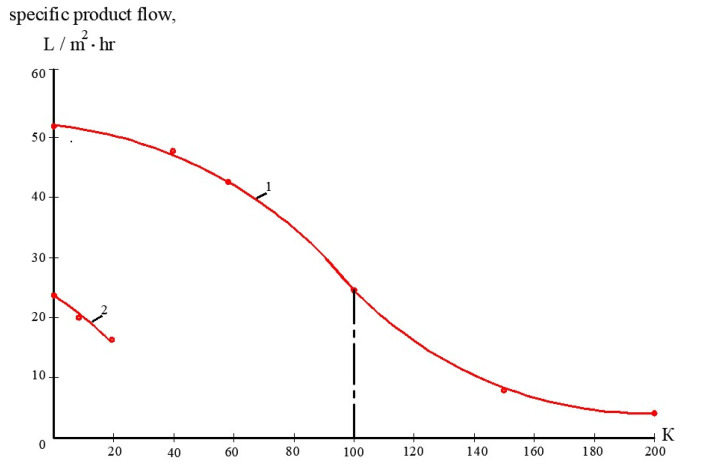
Reduction in nanofiltration and reverse osmosis membrane specific product flow as a function of K in different experiments: 1—reduction in nanofiltration membrane flux in the first series when natural water volume was reduced by 200 times; 2—reduction in reverse osmosis membrane flux in the third series when double-stage treatment scheme was applied to treat reject effluent.

**Figure 12 membranes-13-00133-f012:**
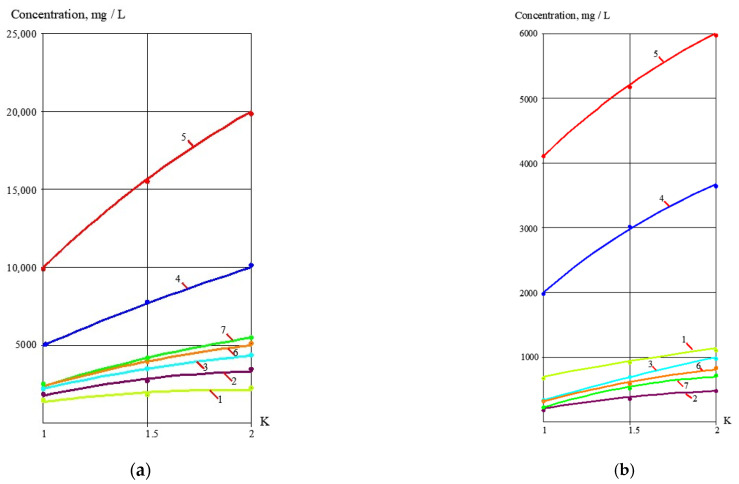
Treatment of reject after sludge dewatering with nanofiltration membranes on the first stage of membrane facility (Figure 2b): (**a**) dependencies of different ion and impurity concentration values on K in concentrate of nanofiltration membranes; (**b**) dependencies of different ion and impurity concentration values on K in nanofiltration membrane permeate: 1—chlorides; 2—sulphates; 3—calcium; 4—bicarbonates; 5—TDS; 6—COD; 7—aluminum.

**Figure 13 membranes-13-00133-f013:**
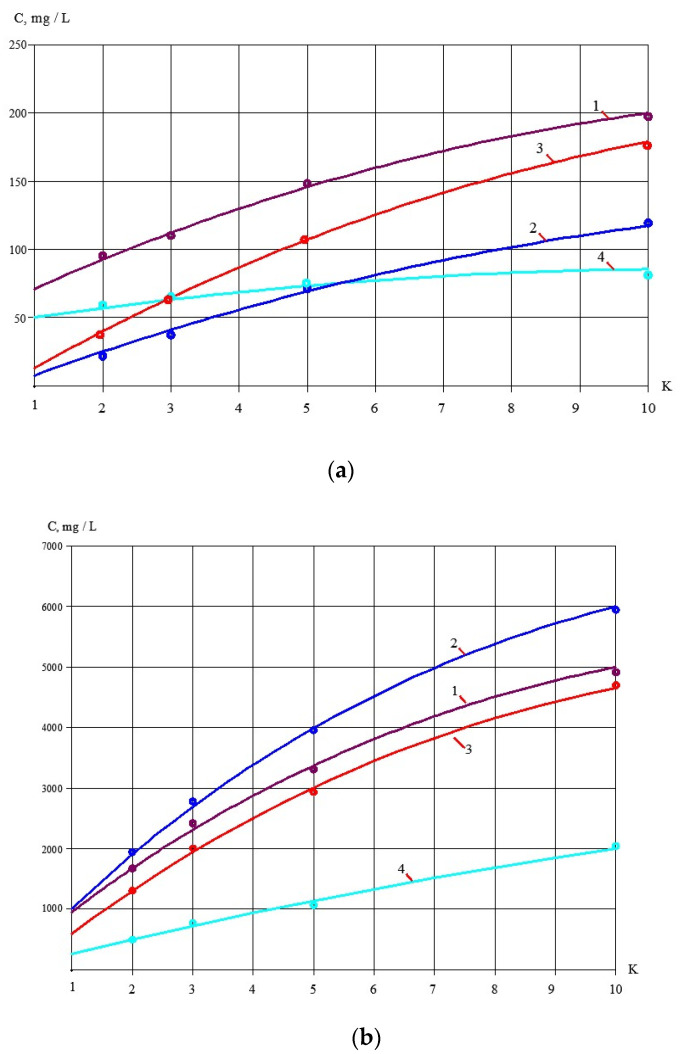
Treatment of the nanofiltration membrane product water by the second-stage reverse osmosis membrane to produce a quality drinking water (Figure 2b) from the reject effluent: (**a**) dependencies of different ion and COD concentration values on K in concentrate; (**b**) dependencies of different ion and COD concentration values on K in permeate: 1—chlorides; 2—sulphates; 3—COD; 4—aluminum.

## Data Availability

The data are available in publications.

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
