# Peer review of "Production of Drinking Water with Membranes with Simultaneous Utilization of Concentrate and Reject Effluent after Sludge Dewatering"

_membranes, 2023, doi:10.3390/membranes13020133_

Round 1

Reviewer 1 Report

1. Figure 2: Description of item 4 is missing. The two streams (permeate and reject) from the RO module are shown on the same side of the membrane, and this may be confusing.

2. The RO permeate in Figure 2(b) has a TDS value of 500 mg/L while the one in Figure 2(a) has a TDS value of 0. Though this is not be wrong based on material balance, so different permeate quality may not be realistic considering that both are RO membranes.

3. The authors may want to check grammatical errors and inappropriate spacing throughout the manuscript (e.g., "on Figure 2(b)", "consist in", "aluminum  and COD", "membrane module3.In membrane" etc).

4. The experimental part (from line 208 onwards) is redundant. Please try to make it short and clear. 

5. In Section 1, the authors provided a lot of details of the principles of utilizing the concentrate and reject effluent. For example, low rejection NF membranes are presented in Figure 3 while combination of NF and RO is presented in Figure 4. However, the authors only showed their experiment of using 1 stage RO membrane (as shown in Figure 5). Details of the 2nd step, which might be more important to this work, are not presented clearly. One thing may confusing the readers is that the authors mentioned NF membrane for the test but RO was used to treat the two liters concentrate (lines 290-292). The authors may want to clarify when and where the NF membrane was used to obtain the results shown in Figure 8. 

Author Response

Dear reviewer,

I would like to thank you for your valuable comments which helped to improve the quality of our article. Please find a detailed description of the comments and their consideration in the article below.

Comment 1: Figure 2: Description of item 4 is missing. The two streams (permeate and reject) from the RO module are shown on the same side of the membrane, and this may be confusing.

Reply to the comment 1: Authors apologize for confusing drawing and have present a new correct one.

Comment 2: The RO permeate in Figure 2(b) has a TDS value of 500 mg/L while the one in Figure 2(a) has a TDS value of 0. Though this is not be wrong based on material balance, so different permeate quality may not be realistic considering that both are RO membranes.

Reply to comment 2: Authors thank the Reviewer for this comment. We made a change in the drawing: permeate TDS of RO membrane is 100 ppm. We used "zero" value for the simplicity when material balance was described.

Comment 3: The authors may want to check grammatical errors and inappropriate spacing throughout the manuscript (e.g., "on Figure 2(b)", "consist in", "aluminum and COD", "membrane module3.In membrane" etc.).

Reply to comment 3: We accept these comments with appreciation and have made necessary corrections.

Comment 4: The experimental part (from line 208 onwards) is redundant. Please try to make it short and clear.

Reply to comment 4: Authors agree and appreciate careful reading. We have removed the excess text.

Comment 5: In Section 1, the authors provided a lot of details of the principles of utilizing the concentrate and reject effluent. For example, low rejection NF membranes are presented in Figure 3 while combination of NF and RO is presented in Figure 4. However, the authors only showed their experiment of using 1 stage RO membrane (as shown in Figure 5). Details of the 2nd step, which might be more important to this work, are not presented clearly. One thing may confusing the readers is that the authors mentioned NF membrane for the test but RO was used to treat the two liters concentrate (lines 290-292). The authors may want to clarify when and where the NF membrane was used to obtain the results shown in Figure 8.

Reply to comment 5: Authors thank the Reviewer for this valuable Comment and ready to provide an explanation. As we had to treat and concentrate large amount of water, to save time and resources only two liters of concentrate were treated on a last stage. It is neither surprising, nor confusing as both pump and membrane pressure vessel have small volume (less than 350 milliliters) and 2-3 liters is a volume usually used in majority of our experiments. We have attached a photo of the laboratory membrane test unit.

Reviewer 2 Report

Please estimate the cost per m3 of the drinking  water produced. NF and RO require high pressures and can be high energy demanding. Compare the above cost with the cost of conventional drinking water production. Discuss the technical and economic viability of your proposal. Main concern: economic viability.

Please update the references since most of them are older than 5 years. Moreover, the only  4 recent references of the paper are self-cites.

Abstract: “Concentrate of membrane plant is mixed with the wet sludge and the 13 reject effluent after sludge dewatering is again treated by reverse osmosis membranes and returned 14 back to the sludge thickening tank” needs and between dewatering and is “sludge dewatering and is again”

¿Have you considered water remineralisation after RO?

Please provide the composition of inlet feed water that corresponds to fig 6.

Discussion of experimental results include 7 figures which are barely discussed. Please relate your results to previous works in the literature.

Author Response

Dear reviewer,

I would like to thank you for your valuable comments which helped to improve the quality of our article. Please find a detailed description of the comments and their consideration in the article below.

Comment 1: Please estimate the cost per m3 of the drinking water produced. NF and RO require high pressures and can be high energy demanding. Compare the above cost with the cost of conventional drinking water production. Discuss the technical and economic viability of your proposal. Main concern: economic viability.

Reply to the comment 1: Authors are grateful for this valuable comment. In this article we did not plan to provide economical explanation and addition of tables. But we add some considerations. Nanofiltration application has definitely reasonable grounds and have economical background. There are a number of projects where standard clarification/filtration approach does not provide quality water. These cases are: high permanganate oxidity (that corresponds to presence of volatile organics, halogenocarbons, pesticides etc.). It is well known that the coagulation does not provide the required product water quality. In these cases ozonation and sorption techniques are used. It is well known and already confirmed in many publications that nanofiltration obviously is less expensive as activated carbon requires high operational costs. This is well discussed in a number of publications, as we made references to a Paris example. We refrain from presenting a table as electricity, activated carbon prices can be very different. The use of NF is obviously less expensive except the payments for concentrate discharge to the sanitary sewer. Our goal was to present a new approach developed by authors to "hide" concentrate in the water withdrawn with the sludge. In Russia the cost of powder activated carbon is 5-6 USD per kilo, 5000 USD per ton. If the powder sorbent dosage is 10 mg per liter, for the water treatment plant with 1000 cubic meters per hour capacity the daily consumption of activated carbon will be 240 kilograms per day and annual carbon consumption will be 70 tons per year and carbon consumption annual costs will be 350000 USD per year. For membrane water treatment plant annual membrane replacement costs (replacement of 1200 membrane modules of 8040 standard) could be less than 100000 USD per year. We do not take power costs that are usually 2-3 times less than membrane replacement costs, also we do not take power consumption of ozonation station that depends on the feed water colour and permanganate oxidity and also could be close to power consumption by low pressure membrane facilities. Also antiscalant (dose of 1-2 ppm) and cleaning chemicals consumption is 3-4 times less than activated carbon costs.

We have added an abstract devoted to evaluation of economical effect:

The described approach to utilize concentrate provides a substantial economic effect. It is well known that to treat water from surface sources with high permanganate oxidity values (over 10-12 mg/l) ozone and sorption techniques are applied. Addition of powder activated carbon of filtration through granular activated carbon bed results in high operational costs as activated carbon has a limited adsorption ability and high cost that varies from 5000 to 10000 USD per one ton of the product. Nanofiltration membranes reject volatile organics, halogenocarbons and pesticides (with equivalent weight that vary from 100 to 300 grams per equivalent) without the use of consumables. For comparison of cost of materials purchased during one year of operation of water treatment station with 1000 cubic meter per hour capacity, we evaluate annual cost of activated carbon dosed 10 mg per liter ranging from 350000 to 700000 USD and annual costs for membrane replacement as 100000-120000 USD. One of the main economical reasons to refrain from using membranes in water treatment practice is the existence of concentrate discharges that can equal to 15-30 per cent of the total feed water flow. Usually, as it is presented on Figure the 2(b), conventional water treatment plants discharge into the sanitary sewer about 2 per cent of the total feed water flow as a reject (or fugate) after sludge dewatering. For the example discussed in the present article and shown on the Figure 2(b), for 1000 cubic meters per hour capacity, assuming the discharge payment of 0.5 USD per one cubic meter, the annual payment for discharge of fugate can be about 70000 USD. This amount is much less than the annual payment for membrane replacement of the membrane facility used to treat and concentrate fugate and to provide concentrate flow reduction from 100 to 4 cubic meters per hour which can be estimated as 12000 USD per year.

Comment 2: Please update the references since most of them are older than 5 years. Moreover, the only 4 recent references of the paper are self-cites.

Reply to the comment 2: Yes, we agree and put eight new references. In fairness, it should be also noted that it was not easy to find new "fresh" publications of research results, devoted to membrane applications to treat surface water and to improve sludge dewatering process. We added new references:

  1. Hao Guo, Wulin Yang, Xianhui Li, Zhikan Yao. Nanofiltration for drinking water treatment: a review. Frontiers of Chemical Science and Engineering, 2021, 15 (5519) DOI: 10.1007/s11705-021-2103-5
  2. Shi Li, Xiao Wang, Yuyue Guo, Jiwen Hu, Shudong Lin, Yuanyan Tu, Lihui Chen, Yonghao Ni, Liulian Huang. Recent advances on cellulose-based nanofiltration membranes and their applications in drinking water purification: A review. Journal of Cleaner Production, Volume 333, 2022,130171
  3. Shahryar Jafarinejad and Milad Rabbani Esfahani. A Review on the Nanofiltration Process for Treating Wastewaters from the Petroleum Industry. Separations 2021, 8(11) 206; https://doi.org/10.3390/separations8110206
  4. Mostafa Hedayatipour, Neemat Jaafarzadeh. Removal optimization of heavy metals from effluent of sludge dewatering process in oil and gas well drilling by nanofiltration. Journal of Environmental Management 203 (Pt 1):151-156; DOI:10.1016/j.jenvman.2017.07.070
  5. Shams Forruque Ahmed, Adiba Momtahin, Fatema Mehejabin, Nuzaba Tasannum. Strategies to improve membrane performance in wastewater treatment. Chemosphere 306(5):135527; DOI:10.1016/j.chemosphere.2022.135527 .
  6. Nikolay Voutckov. Overview of seawater concentrate disposal alternatives. Desalination 273 (2011) 205-219
  7. Ahmed Alghamdi. Recycling of Reverse Osmosis (RO) Reject Streams in Brackish Water Desalination Plants Using Fixed Bed Column Softener. 2017, Energy Procedia 107:205-211, DOI:10.1016/j.egypro.2016.12.174
  8. Noura Najid, Soukaina Fellaou, Sanaa Kouzbour, Bouchaib Gourich, Alejandro Ruiz-Garcia. Energy and environmental issues of seawater reverse osmosis desalination considering boron rejection: A comprehensive review and a case study of exergy analysis. Process Safety and Environmental Protection, volume 156, December 2021, Pages 373 -390.

Comment 3: Abstract: “Concentrate of membrane plant is mixed with the wet sludge and the 13 reject effluent after sludge dewatering is again treated by reverse osmosis membranes and returned 14 back to the sludge thickening tank” needs and between dewatering and is “sludge dewatering and is again”.

Reply to the comment 3: We changed this sentence in the abstract:

Concentrate of membrane treatment plant is mixed with the wet sludge in the thickening tank. The sludge after the thickening tank is dewatered using either filter-press or centrifugal equipment. The reject (or fugate) after sludge dewatering is treated by membrane facility to separate it into deionized water stream and concentrate stream. The deionized water can be mixed with the feed water or drinking water and concentrate stream is returned back to the thickening tank. Thus the salt balance is maintained in the thickening tank whereby all dissolved salts and impurities that are rejected by membranes are collected in the thickening tank and then are withdrawn together with the dewatered sludge.

Comment 4: Have you considered water demineralization after RO?

Reply to the comment 4: We do not discuss what should be done with the product water. It depends on the choice of membrane plant operator. Usually, in water supply business operators aim to return fugate after treatment to the inlet, to mix with the feed water. In majority of cases they fail to clean the fugate, to be more correct, they can reach satisfying results using flocculation techniques, but usually these results are not stable that does not permit to mix this water with drinking water. In our experiments we have demonstrated the possibility to reach high quality water with RO. This water stream can be mixed either with feed water or with product drinking water, or even used for technical purposes such as boiler feed. There is no reason to use remineralization as this stream is small compared with feed water stream thus mixing does not significantly influence  the feed water or drinking water composition.

Comment 5: Please provide the composition of inlet feed water that corresponds to fig 6.

Reply to the comment 5: The composition of the feed water on the Figure 6(b) exactly corresponds to the composition of concentrate shown on Figure 6(a).

We added the following:

Concentrate after the first stage was forwarded to further volume reduction stage. Concentrate composition (concentration values of calcium, chloride, sulphate) as well as COD values in the end of the first series are shown on the Figure 6 (a) and correspond to K value that equals 6. This concentrate was further treated in the second series and was used as feed water. The values of concentrations of all ingredients are shown on Figure 6(b) as functions of K.

Comment 6: Discussion of experimental results include 7 figures which are barely discussed. Please relate your results to previous works in the literature.

Reply to the comment 6: We refrained from long discussion as experimental results are new and goals of experiments were discussed in greater detail. Previous works treated fugates produced after sludge dewatering with membranes [15,16], but the idea to "hide" concentrate in the dewatered sludge belongs to the authors.

We added the following to Discussion:

"Figures 7-10 demonstrate that it can be reached as we collect all impurities and put them into the water that is withdrawn with the dewatered sludge".

Reviewer 3 Report

In this study the authors present an approach that allows to completely exclude liquid discharges during the production of potable water from surface sources. The approach involves reducing the concentrate flow and mixing it with the wet sludge. The study does not present a technological innovation, however it presents important and interesting data.

I suggest some modifications and minor corrections before publication.

Line 30: wasterwater

Line 33: Due to the wide use or capacity or ability

Figures 1, 2, 3, 4 show the flowcharts of the authors' approaches to the water and effluent treatment system, however they are confused to understand. This makes it difficult to understand the process. I recommend lines with different colors or the use of a curve (by-pass) at the intersection of the lines.

I recommend using a table with global rejection values of the contaminants analyzed during the operation.

What would be the cost or gain of using this approach?

Author Response

Dear reviewer,

I would like to thank you for your valuable comments which helped to improve the quality of our article. Please find a detailed description of the comments and their consideration in the article below.

Comment 1: Line 30 - wastewater.

Reply to the comment 1: Authors thank the Reviewer. The correction is made.

Comment 2: Line 33 - Due to the wide use or capacity or ability.

Reply to Comment 2: Authors thank the Reviewer. The correction is made.

Comment 3: Figures 1,2,3,4 are confused to understand. The use of curve (by-pass) at the intersection of the lines is recommended.

Reply to Comment 3: Authors thank the Reviewer. The drawings 1-  are redone and by-pass curves at the intersections are added.

Comment 4: The Reviewer recommends a table with global rejection values of contaminants.

Reply to Comment 4: Authors thank the Reviewer for the comment. But we decided not to include the table, as we have only 5 contaminants and their rejection values are determined throughout the described test runs. The Figure 7 gives the rejection characteristics of membranes for all impurities.

Comment 5: The Reviewer suggests to provide "Cost or gain of using this approach".

Reply to Comment 5: There are two main items that provide evaluation of economic effect of the new approach:

- payment for reagents and service materials;

- payment for wastewater discharges into sanitation sewer;

It is well known that conventional coagulation/sedimentation/filtration techniques produce wet sludge. In many cases when the water treatment plant does not use sludge dewatering, the wet sludge is forwarded to the sanitation sewer. In cases when sludge dewatering devices are used (filter-press or centrifugation dewatering), only fugate after dewatering is forwarded to the sewer that usually is no more than 2 per cent of all feed water treated by the water treatment plant. It is shown on our flow diagram presented on Figure. Another water amount (about 0.4-0.5 per cent of feed water) is withdrawn with the dewatered sludge as the sludge moisture. In Russia the payment for 1 cubic meter discharge to the sewer is 0.5 USD. This fact is often a cause not to use membrane techniques to escape excessive discharges to the sewer as reverse osmosis and nanofiltration facilities usually discharge concentrate that 15-25 per cent.

We added an abstract:

The described approach to utilize concentrate provides a substantial economic effect. It is well known that to treat water from surface sources with high permanganate oxidity values (over 10-12 mg/l) ozone and sorption techniques are applied. Addition of powder activated carbon of filtration through granular activated carbon bed results in high operational costs as activated carbon has a limited adsorption ability and high cost that varies from 5000 to 10000 USD per one ton of the product. Nanofiltration membranes reject volatile organics, halogenocarbons and pesticides (with equivalent weight that vary from 100 to 300 grams per equivalent) without the use of consumables. For comparison of cost of materials purchased during one year of operation of water treatment station with 1000 cubic meter per hour capacity, we evaluate annual cost of activated carbon dosed 10 mg per liter ranging from 350000 to 700000 USD and annual costs for membrane replacement as 100000-120000 USD. One of the main economical reasons to refrain from using membranes in water treatment practice is the existence of concentrate discharges that can equal to 15-30 per cent of the total feed water flow. Usually, as it is presented on Figure the 2(b), conventional water treatment plants discharge into the sanitary sewer about 2 per cent of the total feed water flow as a reject (or fugate) after sludge dewatering. For the example discussed in the present article and shown on the Figure 2(b), for 1000 cubic meters per hour capacity, assuming the discharge payment of 0.5 USD per one cubic meter, the annual payment for discharge of fugate can be about 70000 USD. This amount is much less than the annual payment for membrane replacement of the membrane facility used to treat and concentrate fugate and to provide concentrate flow reduction from 100 to 4 cubic meters per hour which can be estimated as 12000 USD per year.

Round 2

Reviewer 1 Report

The authors may want to study the energy consumption with the additional membrane unit.